# The Effects of Driving Experience on the P300 Event-Related Potential during the Perception of Traffic Scenes

**DOI:** 10.3390/ijerph181910396

**Published:** 2021-10-02

**Authors:** Keiichiro Inagaki, Nobuhiko Wagatsuma, Sou Nobukawa

**Affiliations:** 1College of Engineering, Chubu University, 1200 Matsumoto, Kasugai 487-8501, Japan; 2Faculty of Science, Toho University, Miyama 2-2-1, Funabashi 274-8510, Japan; nwagatsuma@is.sci.toho-u.ac.jp; 3Department of Computer Science, Chiba Institute of Technology, Tsudanuma 2-17-1, Narashino 275-0016, Japan; nobukawa@cs.it-chiba.ac.jp

**Keywords:** driving, driving experience, visual attention, EEG, scene perception

## Abstract

The incidence of human-error-related traffic collisions is markedly reduced among drivers who have few years of driving experience compared with those with little driving experience or fewer driving opportunities, even if they have a driver’s license. This study analyzes the effect of driving experience on the perception of the traffic scenes through electroencephalograms (EEGs). Primarily, we focused on visual attention during driving, the essential visual function in the visual search and human gaze, and evaluated the P300, which is involved in attention, to explore the effect of driving experience on the visual attention of traffic scenes, not for improving visual ability. In the results, the P300 response was observed in both experienced and beginner drivers when they paid visual attention to the visual target. Furthermore, the latency for the peak amplitude of the P300 response among experienced drivers was markedly faster than that in beginner drivers, suggesting that the P300 latency is a piece of crucial information for driving experience on visual attention.

## 1. Introduction

Traffic collisions globally remain a critical social problem [1]. The technology underlying intelligent transportation systems (ITS) has dramatically developed and become widespread in the last few years. As a result, the number of traffic collisions has gradually decreased [2]. Several factors contribute to the occurrence of traffic accidents, including road infrastructure, vehicle systems, pedestrians, and interactions among these factors, including human error [3,4,5]. Human errors, such as misrecognition and poor driving performance, represent a primary underlying cause of traffic collisions [6,7,8,9,10,11]. Most human-error-related traffic collisions are due to errors associated with visual functions, such as less safety confirmation, inattentive driving, misperception, and carelessness [11,12,13,14]. Visual-function-related traffic collisions occur less frequently among experienced drivers with more driving experience than in beginner drivers with few driving opportunities, even if the driver has a valid driver’s license [2,11,15]. Those findings indicate that experience with driving might optimize computational processes of vision, especially visual attention, for the visual perception of traffic scenes. Therefore, understanding the effects of driving experience on the visual attention of traffic scenes is essential for reducing traffic collisions related to human visual function.

A large number of studies on human brain activity during driving using either real vehicles or driving simulators were conducted to understand the effects of distracted driving [16,17,18,19,20], fatigue, drowsiness, and mental workload [21,22,23,24,25,26,27,28] on driving performance [29,30,31]. Brain activity related to driving performance such as acceleration, braking, and reacting to road signs was measured by EEG and fNIRS. Findings correlated driving performance with the powers of the alpha and beta bands [32]. The parietal association cortex and prefrontal area are activated in the recognition and judgment of a traffic environment [33]. Psychoperceptual tasks for the visual perception of traffic scenes show the effects of experience on EEG characteristics associated with visual perception [34,35]. A driver’s brain’s signal processing has gradually been explored in human physical characteristics and driving maneuvers. However, minimal knowledge exists regarding the effects of experience on brain activity during driving [34,35].

Visual attention is a function of the brain that allocates its computational resources to enable us to attend to the most important current information [36] and enhance the visual perception of scenes [37]. Due to this, many studies have been performed to understand the differences in visual behaviors, especially when it comes to visual search, which is very important in visual attention and how it is altered by driving experience. For example, beginner drivers tend to pay more attention to the side lane when compared to experienced drivers [38], who tend to fixate on the location where the vehicle will be in the next few seconds [39,40]. Additionally, more experienced drivers employ a wider horizontal scan area when compared to beginner drivers [34,41,42,43,44,45]. However, the effects of driving experience on signal processing in the brain that underlie visual attention, and the perception of traffic scenes remain largely unexplored, despite large amounts of evidence supporting the relationship between driving experience and visual search [15]. The effect of driving experience on the brain activation pattern in psychoperceptual driving tasks was recently found. Driving experience can be characterized by an EEG activity pattern consisting of alpha and high-beta (gamma) bands [34], and driving experience appears to alter functional connectivity related to gamma-band EEGs [35]. Furthermore, the P300 event-related potential, usually observed at the parietal association cortex, is selectively related to visual attention aiming at multiple targets in a scene, and is a crucial response to understanding the visual-attention characteristics in visual search [46]. These lines of evidence suggest that driving experience might affect brain-signal processing related to visual attention, as reflected by P300 during the visual search of traffic scenes. This is shown by measuring and evaluating P300 during the visual search of traffic scenes.

The present study focuses on visual attention, an essential visual perception and cognitive function, and investigates the relationship between visual perception and driving experience on the basis of EEG recordings and analysis. We focused on event-related potential P300, which is strongly correlated with visual attention [47,48,49,50,51,52,53,54,55]. We evaluated changes in the response characteristics of the P300 to the attentional target during the visual search of multiple targets in traffic scenes, obtained from experienced and less experienced drivers to investigate the effect of driving experience.

## 2. Methods

### 2.1. Subjects

This study recruited 17 healthy subjects, namely, 1 female and 16 males aged 19–26 years, to participate in EEG recording during the visual perception of traffic scenes. All subjects had a valid driver’s license, normal or corrected-to-normal vision (using glasses or contact lenses), and presented with normal EEG patterns. The definition of driving experience is complicated. In this study, to investigate the effects of driving experience, the subjects were divided into experienced and beginner drivers on the basis of their driving frequency and the duration for which they have held a driver’s license [2], which is the same definition as that in our previous study [34,35]. In this study, our experienced subjects had driven more than five times per week and maintained a valid driver’s license for at least three years. In contrast, our beginner subjects had few driving opportunities or had maintained a valid driver’s license for less than one year. The average driving frequency in one week among our subjects was 5.73 ± 0.53 for experienced drivers and 0.32 ± 0.15 for beginner drivers, indicating that our beginner subjects had almost no or minimal driving experience. The average holding duration of a driver’s license was 5.03 ± 0.82 for experienced drivers and 1.06 ± 0.61 for beginner drivers. The subjects were eight experienced drivers and nine beginners. Before the experiment, we asked subjects to determine fatigue and arousal, and those subjects who had reported fatigue or reduced arousal did not participate in the experiments.

### 2.2. Experimental Design

Figure 1 summarizes the custom-designed experimental apparatus used to measure EEGs during the perception of driving scenes. The experimental apparatus included a chin rest, a driver’s seat, a personal-computer (PC) display (1920 × 1080), and a PC utilized for stimulus control and EEG acquisition. During the experiment, several traffic scenes th had been captured using a video camcorder were projected onto the PC display (1920 × 1080). Driving skill was associated with the traffic scene; therefore, we used several types of traffic scenes from urban areas. Variability among the number of possible attentional targets, such as traffic signs, pedestrians, and other vehicles, was similar across scenes. The distance between the projected PC display and the subject was maintained at 60 cm, which simulated the experience of watching traffic scenes from the driver’s seat in a vehicle. Under the experimental conditions, the field of view was ±30°, which encompassed the functional field of view during driving (approximately ±20°) [56].

### 2.3. Experimental Stimulus and Protocol

#### 2.3.1. Stimulus

Figure 2 summarizes the experimental procedures used to test the visual perception of traffic scenes. In this study, traffic scenes were used for the visual search task because of multiple traffic targets such as road signs, other vehicles, and pedestrians. In addition, the experience of driving is relatively easily evaluated because driving establishes driving experience in daily life. Traffic scenes were recorded in an urban area with 1280 × 720 resolution using a video recorder (Sony, 30 fps) mounted on the dashboard in front of the driver, so as not to block the driver’s view, according to traffic safety regulations in Japan.

In the experiment, we used two types of stimulus scenarios: a stimulus trial that included a visual target (green circle shown in Figure 1 and Figure 2) in the traffic scene, and a no-stimulus trial that was a typical traffic scene without a visual target. During the stimulus trial, a visual target appeared for 150 ms at random timing in the left hemisphere of the traffic scene due to drivers in Japan driving in the left lane. The size and color of the visual target were set to be 1.5° and green, respectively. Each trial consisted of presenting the traffic scene for 10 s with 5 s rest intervals, during which a gray image was displayed. The subjects viewed those two types of stimulus scenarios from the perspective of an individual driving a vehicle. They perceived the traffic target the same as when they were ordinarily driving, and perceived visual targets that appeared at unexpected times. In the experiment, 30 stimulus and 40 no-stimulus trials were randomly presented to the subjects for the stimulus trial to be an oddball task.

#### 2.3.2. Experimental Protocol

We first explained the experimental design to the subjects and obtained informed consent. Then, the EEGs of all subjects were measured during the visual perception task described above to assess their responses to traffic scenes. Subjects sat in the driver’s seat and were allowed to adjust it for their comfort. The PC display was adjusted to a similar visual angle, reflective of normal driving. The experiment was performed under silent conditions to avoid auditory interference. During the experiment, we asked the subjects to avoid body movements to reduce artifacts. After watching each traffic scene, we also asked the participants about their arousal level.

During the experiment, we instructed the subjects to watch the traffic scene as though they were engaged in everyday driving, and informed them that a visual target might appear at a specific scene position to capture the subject’s attention and gaze. We also instructed subjects to react to the appearance of the visual target by pressing the up button on the keyboard. The Ethics Committee of Chubu University approved the study protocol (20210016). We performed all methods in accordance with the Declaration of Helsinki.

### 2.4. EEG Recording

EEGs of all subjects were measured using a Polymate AP108 (Miyuki Giken Co., Ltd., Tokyo, Japan). To record EEG data, electrodes were placed according to the International 10–20 electrode positioning system [57]. Before electrode placement, the scalp of each subject was cleaned using alcohol to avoid EEG signal deterioration due to scalp oil. Sampling frequency was 500 Hz. We used EEG signals measured from the Cz electrode, located at the parietal area of the brain, to analyze the P300 event-related potential [48]. During the EEG recording, each subject placed their head on a chin rest to reduce noise caused by body and head movement.

### 2.5. Analysis

P300 event-related potential was determined by averaging 20–30 EEG responses to low-frequency stimuli in an oddball task organized according to low- and high-frequency stimuli [47,48]. In our experiment, we considered the response to the visual target to be the low-frequency stimulus, and averaged these responses to determine P300.

At first, the EEG signal in both stimulus and no-stimulus trials was filtered into the delta band (1–5 Hz) using a band-pass filter because P300 most commonly occurs in the delta band. To calculate the average response for the stimulus trial, the filtered EEG between the pre- (−200 ms) and postappearance of the visual target (800 ms) was selected for all stimulus trials and averaged. We also calculated the average EEG response for the no-stimulus task by randomly selecting 1 s of EEG data for each trial, averaged together. When calculating averages, data exceeding three times the standard deviation, computed for each trial, were removed as likely interference due to noise caused by body movements or eye blinks.

Several definitions are used to describe the P300 event-related response [48], which can vary depending on stimulus modality, task conditions, and age. The EEG typically displays a peak response between 250 and 600 ms following stimulus presentation, defined as P300. We defined the P300 response as the peak response identified between 250 and 600 ms after the appearance of the visual target in our study.

Several functions, such as attention, interest in the stimulus, and concentration, are reflected in P300 characteristics, including peak amplitude and latency to peak amplitude [47,48,49,50,51,52,53,54,55], which we computed in our study. Peak amplitude for the P300 was calculated using the baseline-to-peak method. The baseline was defined as the average amplitude between −200 and 0 ms, relative to the appearance of the visual target. Peak amplitude was calculated by subtracting the baseline amplitude from the P300 peak value between 250 and 600 ms after the appearance of the visual target. Latency to peak amplitude was defined as the time between the appearance of P300 following the visual-target presentation.

The perception rate for the visual stimulus that appeared in the stimulus trials was calculated by the number of buttons pressed for the visual target divided by the total number of visual target appearances.

## 3. Results

### 3.1. P300 Response

We calculated the average P300 event-related potential from beginner and experienced drivers. Figure 3A,B show the average for the observed P300 event-related potentials in beginner and experienced drivers during the stimulus trials, and the averaged response during the no-stimulus trials, respectively. During stimulus trials, the P300 event-related potential was present with a peak response between 250 and 600 ms after the appearance of the visual stimulus in both experienced and beginner drivers. P300 was observed in all experienced drivers, whereas this response occurred in 8 of 9 beginner drivers. One beginner driver did not present any P300 response, and the EEG obtained from this subject was not used further evaluation of P300 characteristics. During the no-stimulus trial, no clear peak P300 response was observed for either beginner or experienced drivers.

### 3.2. P300 Characteristics

P300 latency and peak amplitude reflect the effects of attention, concentration, and interest in a task. We analyzed the latency and peak amplitude of P300 for beginner and experienced drivers during the perception of traffic scenes. Figure 4 summarizes the latencies and peak amplitudes of beginner and experienced drivers. First, P300 latencies in beginner and experienced drivers were 0.540 ± 0.061 and 0.451 ± 0.030 s, indicating that P300 latency was faster among those with driving experience. We compared the latency values using the Wilcoxon rank-sum test and confirmed that the difference between groups was significant (*p* < 0.01; experienced drivers = 8, beginners = 8).

We also evaluated peak P300 amplitudes for beginner and experienced drivers during the perception of traffic scenes. A slightly larger peak amplitude was observed for P300 with experienced drivers than that for beginner drivers (beginners: 8.13 ± 2.01 mv, experienced drivers: 8.82 ± 1.73 mv). However, there was no significant difference after comparison using the Wilcoxon rank-sum test.

### 3.3. Response to Visual Stimulus

Table 1 summarizes the response to the visual stimulus among beginner and experienced drivers during the experiment. As shown in Table 1, there was high visual stimulus perception for both beginner and experienced drivers (beginners: 94.1%, experienced drivers: 96.3%). This result indicates that both beginner and experienced drivers could perceive the visual target used in our experiment, because the subjects were given instruction to pay attention to the target before the experiment, even when several targets such as road signs, pedestrians, and other vehicles occurred in the traffic scenes.

## 4. Discussion

Our study focused on the P300 event-related potential, which is associated with attention [47,48,49,50,51,52,53,54,55], and investigated whether P300 could be used as a measure of brain activity in a driver reflecting driving experience during the perception of traffic scenes and the response to a stimulus. In our experimental protocol, we informed the subjects of the appearance, position, and characteristics of the visual target projected onto the traffic scene to ensure that the subjects paid attention to the stimulus and reduced variability. As a result, the perception rate for the visual target was high among both experienced and beginner drivers, indicating that experienced and beginner drivers paid attention to the visual target. Under this attention-concentrating condition, P300 was observed in both beginner and experienced drivers. These findings indicate that P300 is highly induced during EEG recordings when the driver focuses visual attention on a visual target.

In prior studies, the number of traffic collisions related to human error decreased as driving experience increased [11,15]. Furthermore, human-related traffic collisions are more common among drivers with minimal driving experience, even if they have valid driver’s licenses, such as newly licensed drivers with less than one year of experience. Among beginner drivers and drivers with limited driving experience, traffic collisions are often caused by visual functions, such as less attention paid to targets involved in the collision or misperceptions. As a result, higher traffic-collision rates are reported for beginner drivers compared with for experienced drivers. We recently reported that the EEG activation patterns of drivers are uniquely altered between experienced and beginner drivers [34]. Moreover, the estimated gamma-band functional connectivity from EEG recordings of the whole-brain area is entirely different between experienced and beginner drivers [35]. These lines of evidence imply that driving experience may optimize the visual-signal-processing functions that underlie visual cognition and perception during driving. In our experiments, bias toward visual attention to the appearance of a target in the traffic scene was strongly induced by the provided instructions, resulting in a high rate of P300 detection in both experienced and beginner drivers. However, the characteristics of P300 differed between groups, as latency to peak P300 amplitude was significantly delayed in beginner drivers compared with experienced drivers, although no significant difference was appeared in peak P300 amplitude. These findings indicate that the effects of driving experience on visual attention or perception reflected latency to peak P300 amplitude rather than to peak P300 amplitude.

In our experiment, participants’ driving experience was associated with latency to peak P300 amplitude and not peak P300 amplitude. P300 can be divided into two components, P3a and P3b, on the basis of latency [47,48]. In general, P3a has earlier latency than P3b does. P3a and P3b code different types of information, with P3a strongly associated with attention, and P3b more related to task concentration [47,48]. In our results, as shown in Figure 3, the P300 response was difficult to separate into P3a and P3b components. This may indicate that P300 measured in our task had both P3a- and P3b-related components, and both were altered by driving experience. Consequently, changes in the balance of P3a and P3b could be manifested as changes in P300 latency in our experimental results. A decrease in alpha-band activity and the induction of beta- or gamma-band activity are widely accepted as being associated with concentration [58,59,60,61]. As we reported previously [34], beginner drivers’ beta- and gamma-band spectral responses were higher than those among experienced drivers, whereas alpha-band activity in beginner drivers is reduced compared with in experienced drivers. Our findings suggested that beginner drivers were more likely to concentrate on a traffic target needed to be perceived rather than paying attention to the entire traffic scene, whereas experienced drivers had the opposite response. This increased in the task concentration-related P3b component of P300, reflected by the slow latency to peak amplitude among beginner drivers. By contrast, a faster P3a component of P300 was observed among experienced drivers. On the basis of the relationship between P3a and attention, our results imply that experienced drivers might appropriately apply visual attention to the perception of traffic scenes compared with beginner drivers, resulting in a faster P300 response. From another point of view, reaction time for visual targets may also vary according to driving experience. Reaction time for a visual stimulus may be faster by driving experience, and this may correlate with our findings, with the P300 response being faster with experience. This might be a critical finding, which may couple attention-related brain activity to the perceptual reaction for a target, and we will investigate it soon.

In the present study, we evaluated the EEG response of only one recording position (Cz). To date, many studies have used EEG to understand driving-related human properties, such as fatigue, drowsiness, and the effects of the distractors, which typically involves analysis of the power spectrum and wavelet transformation. More recently, inter-regional neuronal activity has been evaluated as an indicator of the strength of functional connectivity to understand associated functions with the activity of the whole-brain network [62,63,64,65,66,67]. Brain-signal processing related to visual perception and recognition is often hierarchically and widely spread throughout the brain. For example, we previously reported that the gamma-band functional connectivity estimated from whole-brain EEG recordings is significantly different between experienced and beginner drivers [35]. To further understand the signal processing performed by the whole-brain network during visual perception during driving and to examine the effects of experience, whole-brain EEG recordings are necessary to investigate the relationship between P300 induction and functional signal processing in the brain network through functional connectivity or coherence analysis with graph theory.

Driving experience alters latency to the P300 response, although it has almost no effect in its amplitude when using an artificial visual target. P300 amplitude reflects an interest in the stimulus [47,48], and thus the P300 amplitude response for natural targets in traffic scenes such as pedestrians, load signs, and crossing load. To further understand the relationship between P300 and driving experience in the visual perception of traffic scenes by considering the future implementation of our findings as a driver support system, the response characteristics of P300 are validated for natural visual targets in a traffic scene.

## 5. Conclusions

We focused on the relationship between attention and the P300 event-related potential to investigate the effects of driving experience on response characteristics. P300 was highly observed in both experienced and beginner drivers due to the strong induction of visual attention to the target in response to task instructions. P300 latency becomes faster by driving experience. Experienced drivers showed faster P300 responses, while beginner drivers showed slower P300 responses, which may reflect changes in the proportions of the P3a and P3b components in response to concentration on the task or the optimization of visual attention during driving, determined by driving experience. In a future study, we will investigate the P300 responses for original objects in traffic scenes rather than artificial visual targets to understand the essential characteristics of visual perception during driving. By this investigation, the P300 response for original objects in traffic scenes could be quantified, for instance, the effect of location (e.g., foveal or peripheral) and saliency (intensity of the object affected to visual attention). This understanding and quantification for the relationship between visual attention during searching a traffic scene and P300 response characteristics are implementable on a driver support system a vehicle function. We also aim to analyze EEG recordings from multiple brain positions to understand which types of brain-signal processing are associated with the P300 response, and how these vary with changes in driving experience based on the computation of functional connectivity.

## Figures and Tables

**Figure 1 ijerph-18-10396-f001:**
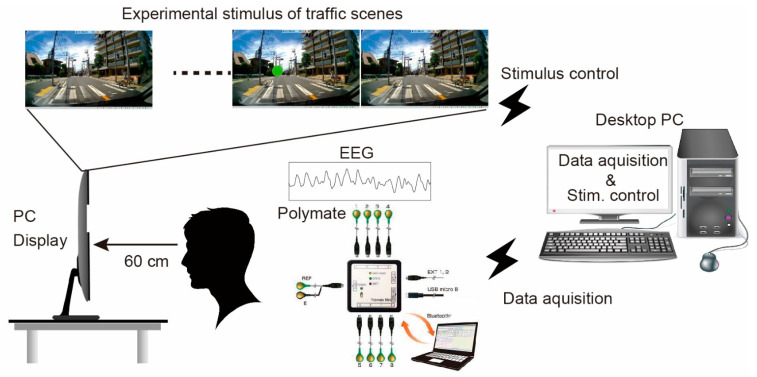
Experimental setup used to test the perception of traffic scenes, consisting of a PC display, adjustable chair, chin rest, and PC. Several daylight traffic scenes were captured using a video camcorder (1280 × 720 pixels, 30 fps) before the experiment, which was shown on the PC display as the visual stimulus from a PC. Polymate AP108 (Miyuki corporation, 500 Hz) was utilized for EEG recording. The recorded EEG was transferred to the PC via Bluetooth connection. Custom-designed MATLAB software run on the same PC controlled the visual stimulus and EEG recording projection.

**Figure 2 ijerph-18-10396-f002:**
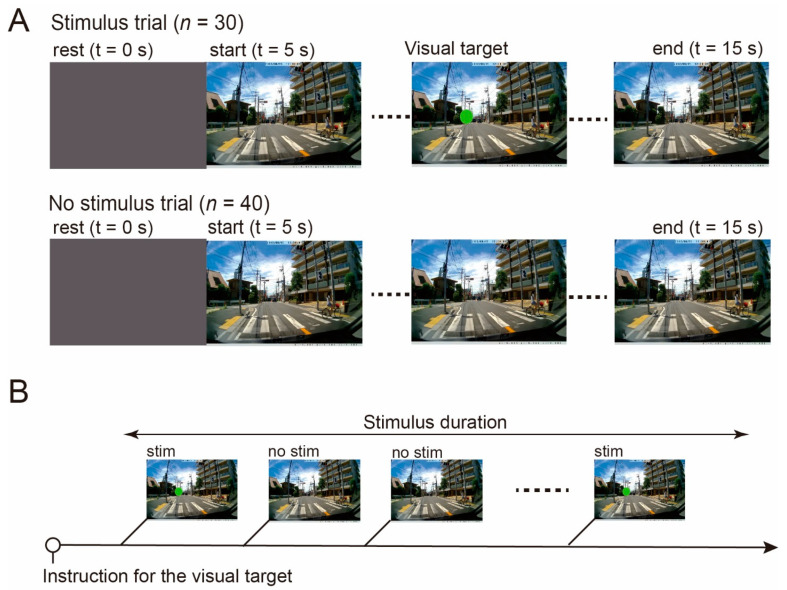
Experimental stimulus of traffic scenes. Subjects were shown several types of traffic scenes during the experiment and told to react to the artificial visual target. (**A**) Experimental trial during which the traffic scene was displayed for 10 s following a 5 s display of a gray stationary image (resting). An artificial visual target was inserted into the traffic scenes consisting of a green circle 1.5° in diameter on the center of the left hemisphere of the visual field. The target appeared at random for 150 msec. (**B**) We used 30 sessions of traffic scenes, including an artificial visual stimulus (stim trial), and 40 sessions of traffic scenes without any stimulus (no-stimulus trial). Each subject saw 70 traffic scenes. Subjects were asked to view the traffic scenes as though they were driving an ordinary vehicle during the entire session. Participants were informed that a visual target would appear in the traffic scene to induce strong visual attention. Stimulus and no-stimulus trials were randomly presented.

**Figure 3 ijerph-18-10396-f003:**
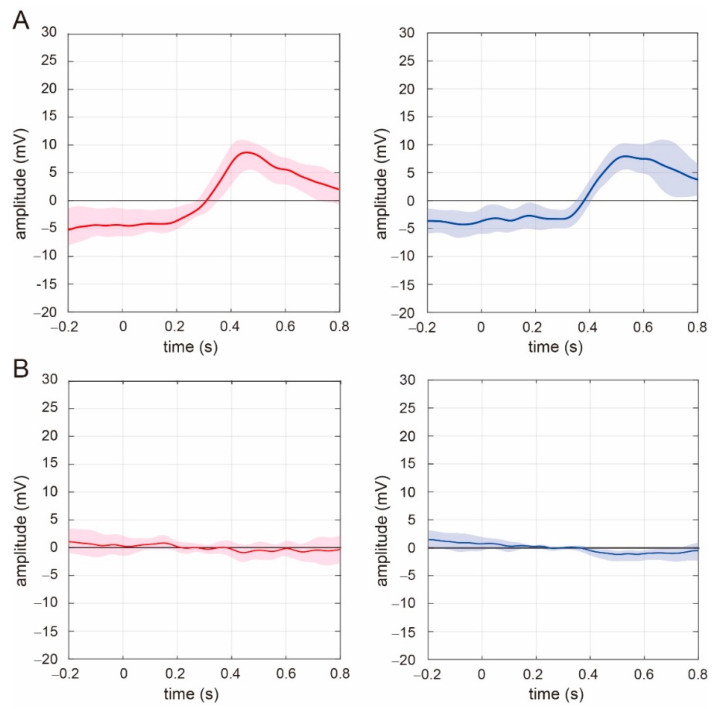
Overall average for event-related potential component P300 among (**left**) experienced and (**right**) beginner drivers during (**A**) stimulus and (**B**) no-stimulus trials. Average waveforms for experienced and beginner drivers calculated from 8 and 8 subjects, respectively. Solid lines and shaded areas indicate average and ±1 SD, respectively.

**Figure 4 ijerph-18-10396-f004:**
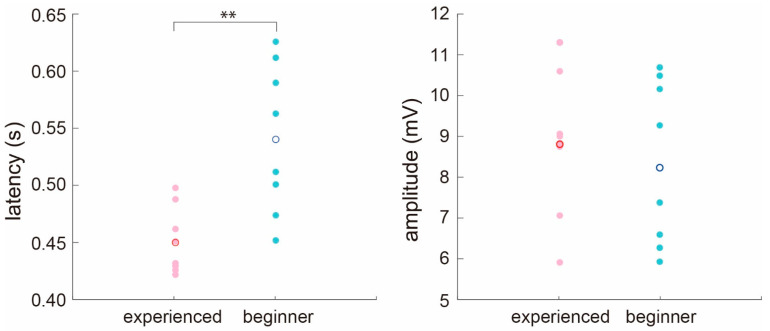
(**left**) Latency for peak amplitude of P300 and (**right**) peak amplitude of P300 in beginner and experienced drivers. Cyan and magenta circles are individual responses of beginner and experienced drivers, respectively, and blue and red circles are average values. **, significance (*p* < 0.01, Experienced drivers = 8, beginners = 8), as evaluated by Wilcoxon rank-sum test.

**Table 1 ijerph-18-10396-t001:** Recorded parameters during visual attention task according to driving experience.

Parameter	Experienced Drivers (*n* = 8)	Beginner Drivers (*n* = 8)
Perception rate for visual target (%), and number of responded targets.	96.3 (231/240)	94.1 (226/240)
Latency (s).	0.451 ± 0.030	0.540 ± 0.061
Peak amplitude (mV).	8.82 ± 1.73	8.13 ± 2.01

## Data Availability

Datasets generated during the current study are available from the corresponding author on reasonable request.

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
