# Peer review of "The Effects of Driving Experience on the P300 Event-Related Potential during the Perception of Traffic Scenes"

_ijerph, 2021, doi:10.3390/ijerph181910396_

Round 1

Reviewer 1 Report

The paper discusses the effects of driving experience on the visual perception of traffic scenes that require attention by testing the latency of peak in P300 response.

The test reference, peak amplitude of P300 and the latency to approach it, is a new way of research on this field. The paper proves the inferior position of inexperienced drivers in another method.

In my opinion, more discussion can be added regarding to the specific implementation of the future work.

Overall, I can accept the paper being published in present form.

Reviewer 2 Report

The abstract should be reformated to put more emphasis on the protocol and its results. The current form is too long in its introduction. 

Line 25, I suggest not using the "We" form through the text. When results are positive, the authors use We, when it is negative or when there is no effect, they use the neutral form, this is not appropriate.

Abstract should finish with an opening or futur works, the last sentence is a summary of results and despite being interesting for research, it is not a conclusion.

The use of accidents has been forbidden for collisions, and the text should do the same.

Line 55, format the "We"

Line 56, it should be specified if the driving refers to actual driving of a simulator or if it is a driving paradigm such as psycho-perceptual task, which are not the same since one does not have to drive in the later one. The same should be done for section of line 72. Line 88 suggests that the protocol is not driving per se.

64: visual search is a more appropriate term instead of eye movements since we are mostly implying, especially with experience that driving is a top-down task

78 to 83: objective is clear but the last sentence is hard to grasp since we don't know if it is an hypothesis or just a statement, this has to be clarified.

Line 100: Were subjects removed due to fatigue ? It should be specified because we can't fin any "n" further down the manuscript or in statistical model.

120: we used presented : there is an unecessary verb.

121 : what is the visual target (stimulus trial) ? is it the green dot appearing on figure 1 and 2, a reference to the task should be made quickly in the manuscript to clarify the procedure used.

line 207, the number of subjects should be clarified upfront in the method where we state 20, but the results only present 17.

Figure 3, how was the no-response novice treated in the average presented of figure 3 since it might influence greatly the results ?

How can figure 4 presents results of 9 beginners if line 213 says  only 8 out of 9 showed P300 ?

Figure 4, I suggest not overlapping individual results to allow the reader to fully appreciate the variation of individual responses.

As suggested above for n presentation, Table 1 should specify the number of participants per group and perception percentage should also present the total number of trials and not only %.

Section 3.3 strongly suggest the stimulus had a bottom-up effect on visual search/identification, this should be described further down the manuscript since it is not as clear why experience behind the wheel would explain such a difference in a static environment where one does not have to drive.

Line 284, I would suggest using "could be" instead of are, the results of the present experience are not that conclusive and using such a strong sentence goes beyond the results obtained among only 17 participants.

Although authors make references to driving experience as for group and the driving scene presented, it is hard to understand what would be the difference if the stimulus would have been just blank screen with the presence of the target (or not). The actual paradigm is not associated with visual search but a go no-go decision based on the green dot appearing in the left hemisphere. This have to be clarify through the text and better linked with results.

Despite being of interest for road safety and research on driving experience, the manuscript have to be modified to better highlight the application of the results obtained and how they transfer to actual driving context and/or interventions.

Reviewer 3 Report

The study aims to investigate the impact of driving experience on P300, and the relationship between P300 and attention while seeking for response differences between experienced and novice drivers. The idea that experience can impact on the P300 amplitude and latency is interesting and can contribute to the existing studies

Abstract

The abstract should be more straight to the point explaining clearly what the research aim is and avoiding irrelevant information not related to the study (e.g. intelligent transport system). The background occupies the majority of the abstract. The authors should summarise the background information and provide more details about the adopted methodology and results.

When the authors refer to “visual functions underlying visual perceptions during vehicle operation due to increased driving experience”, they should make clear what visual functions they refer to.  Do they refer to the ability to perceive and spot a specific information on the traffic scene due to the improvement of drivers’ skills?

In addition, the authors should make clear that they do not refer to an increased ability to see (visual impairment).

Introduction

The introduction is rather scarce and would benefit from more detail and background into the topic. A bit more information about the neuroscience of driving is warranted.

The authors should provide more detail and expand more on the correlation between P300 and visual attention for a better and more robust support of the research question. The introduction should also contain information on how this relationship has been measured so far, specifically in driving related research (e.g. tasks, stimuli), to justify the selected methodology.

In order to fully understand the aim of the study, the introduction should contain more information about the relationship between visual perception and driving experience. This will help justify the chosen task (spotting a target in a timely manner). It should be explained why scanning the location in search of targets is relevant for the chosen study and why this is an indication of experience.

Page 1 Line 38 While it is true that AI has contributed towards road collision decrease, it is not clear to me how this information is relevant for the current study. Perhaps, including statistics on collisions due to driving distraction and inattention will be more useful.

Page 3 Line 47-49. Could you explain what “computational process” do you refer to and how are it is relevant to the task adopted by the current research? Providing more detail about the cognitive processes of interest will make the aim of the study much clearer. 

Method

Driving experience per se is a complex construct and defining it is not a trivial task. Driving experience is one of the main variables in this study, therefore the demographic information regarding driving experience should be more detailed. The authors should provide kilometres/miles driven per week, average months or years of driving licence, average age per group. The differences in experience should be clearly presented. What was the reason to choose as a cut point 3 years for driving experience?

The authors should explain why the experienced drivers are called “expert drivers” throughout the paper. Could we define a driver with only 3 years of driving experience as an “expert”? If the experienced participants do not hold a professional driving licence (or other indication of expertise), I suggest changing “expert” for “experienced” drivers.

Can the authors clarify whether they intended to measure driving skill or they were just interested in observing the changes in P300 according to the differences in driving demographics (years of driving) reported by the participants? If the second is true, then this needs to be clearly explained.

The authors mentioned that they asked participants to react to the target by pressing a button. Do I understand that the authors collected reaction time data? If so, it will be interesting to see whether experienced drivers reacted faster to the target and correlate RT responses with the EEG data. The use of RT data is a common measure for driving skill and experienced drivers react faster to target hazards than novices.

More information should be provided on how the traffic scenes were recorded. Was it during everyday driving? How were the cameras attached? Did the authors participate in the recording of the videos? Were they following a specific filming protocol?

The authors should provide a more detailed description of the driving scenarios. The authors have randomly selected 1 second of EEG data for each trial that did not contain the stimulus. Were both non-stimulus scenarios and those that contain the stimulus similar?   

Can the authors clarify how perception rate was measured? By asking participants to react to the target?

Discussion

Line 291, Page 8 The authors say “These findings suggested that beginner drivers were more likely to concentrate on the perception task rather than paying attention to the entire traffic scene, whereas expert drivers had the opposite response” Does this mean that the experienced drivers were scanning more globally the traffic scenarios instead of focusing on the point where the target would appear?

The results suggest that “difference in the P300 response caused by driving experience was reflected in the latency to the peak P300 amplitude rather than the peak P300 amplitude”. However, what does this mean in relation to driving experience? With the P300 being present in both groups however differing in latency, what is this result telling us about the processing differences between experienced and novice drivers?
